biochemistry

amyloid, Alzheimer disease, oligomerization, aggregation kinetics, neurodegeneration

**Author for correspondence:**
Zhefeng Guo
e-mail: zhefeng@ucla.edu

# Aβ42 fibril formation from predominantly oligomeric samples suggests a link between oligomer heterogeneity and fibril polymorphism

Christine Xue, Joyce Tran, Hongsu Wang, Giovanna Park, Frederick Hsu and Zhefeng Guo

Department of Neurology, Brain Research Institute, Molecular Biology Institute, University of California, 710 Westwood Plaza, Los Angeles, CA 90095, USA

ZG, 0000-0003-1992-7255

Amyloid-β (Aβ) oligomers play a central role in the pathogenesis of Alzheimer's disease. Oligomers of different sizes, morphology and structures have been reported in both *in vivo* and *in vitro* studies, but there is a general lack of understanding about where to place these oligomers in the overall process of Aβ aggregation and fibrillization. Here, we show that Aβ42 spontaneously forms oligomers with a wide range of sizes in the same sample. These Aβ42 samples contain predominantly oligomers, and they quickly form fibrils upon incubation at 37°C. When fractionated using ultrafiltration filters, the samples enriched with smaller oligomers form fibrils at a faster rate than the samples enriched with larger oligomers, with both a shorter lag time and faster fibril growth rate. This observation is independent of Aβ42 batches and hexafluoroisopropanol treatment. Furthermore, the fibrils formed by the samples enriched with larger oligomers are more readily solubilized by epigallocatechin gallate, a main catechin component of green tea. These results suggest that the fibrils formed by larger oligomers may adopt a different structure from fibrils formed by smaller oligomers, pointing to a link between oligomer heterogeneity and fibril polymorphism.

## 1. Introduction

Protein aggregation is involved in a wide range of human diseases [1]. The end aggregation product is often the pathological

hallmark in these disorders. For example, amyloid-β (Aβ) aggregation leads to the formation of senile plaques in Alzheimer's disease [2]; α-synuclein forms Lewy bodies in Parkinson's disease [3]; and the tangles in various tauopathies are the result of tau aggregation [4]. The end aggregation product of these proteins is called amyloid [5], which has well-defined histochemical, biophysical and biochemical characteristics [6,7]. To understand the pathology in these amyloid-related disorders, the process of protein aggregation that leads to amyloid formation has attracted extensive research efforts. At the same time, it has been well established that protein aggregation also leads to the formation of soluble intermediates, which are often called oligomers. These oligomers are shown to be more toxic than the amyloid fibrils in a variety of activity assays using cultured cells and transgenic animals (reviewed in [8]). Some studies suggest that oligomers eventually are converted to amyloid fibrils [9,10] or form the building blocks of amyloid fibrils [11–15], and other studies show that the amyloid fibrils can also promote the formation of oligomers [16]. The relationship between oligomers and amyloid fibrils in the aggregation process is critical for a complete understanding of the underlying aggregation process and for a rational approach to develop therapeutic interventions.

In Alzheimer's disease, the senile plaques are mainly composed of the amyloid fibrils of Aβ protein [2]. Aβ is not a homogeneous species, consisting of two main isoforms: Aβ40 and Aβ42, and other minor truncated and modified Aβ variants. Aβ40 is the most abundant Aβ isoform in the brain [17–19], but Aβ42 is the major Aβ isoform in the amyloid plaques [20–23]. Experimental evidence suggests that Aβ42 and Aβ40 form interlaced amyloid fibrils [24] and they can seed each other's aggregation [25–27]. Aβ fibrillization starts with a nucleation process in which small fibril nuclei are formed [28]. These fibril nuclei adopt the structure of the fibrils and are capable of growth by monomer addition. Fibrils are also capable of promoting nucleation through a secondary nucleation process [16]. Aβ oligomers have been shown to adopt different structures from fibrils [29–31]. The relationship between oligomerization and fibrillization is not fully understood. Some studies suggest a fibrillization model of nucleated conformational conversion [9,10,30]. This model starts with the formation of Aβ oligomers, which then convert to fibril nuclei. There are also studies suggesting that Aβ forms ring-like oligomers, which form the building blocks of amyloid fibrils [15,32,33]. It is unclear how Aβ aggregation proceeds in the brain.

To this end, here we report that Aβ42 aggregation starts with oligomerization. In phosphate-buffered saline (PBS), Aβ42 forms oligomers almost immediately without fibril formation. Monomers account for only a small percentage of the Aβ42 population. Upon incubation at 37°C, the Aβ42 sample with predominantly oligomers proceeds to form fibrils and display sigmoidal aggregation kinetics. We further show that the Aβ42 sample enriched with smaller oligomers form fibrils at a faster rate than Aβ42 sample enriched with larger oligomers. Aβ42 fibrils formed by small and large oligomers appear to have different structural characteristics, suggesting a link between oligomer heterogeneity and fibril polymorphism.

# 2. Results and discussion

## 2.1. Aβ42 quickly forms oligomers with a wide range of sizes

For all the studies in this work, we used recombinant Aβ42 proteins with a protocol that we optimized over the years. The recombinant Aβ42 protein in this work does not contain any extra residues. All the purification steps were performed in high-pH buffers in the presence of chemical denaturants to limit the formation of Aβ aggregates. Following lyophilization, the Aβ42 powder was treated with hexafluoroisopropanol (HFIP). HFIP treatment has become a common practice in Aβ aggregation studies. To study aggregation, Aβ needs to be dissolved in an aqueous buffer, which is often the PBS buffer. For this purpose, we first dissolved Aβ42 protein in a high-pH denaturing buffer containing 7 M guanidine hydrochloride at pH 11. Both high pH and high concentrations of denaturant help inhibit Aβ aggregation. Then we performed a buffer exchange step with a desalting column to transfer Aβ protein to PBS buffer. We checked the morphology of the Aβ sample after buffer exchange using transmission electron microscopy (TEM), which shows that the Aβ42 sample contains abundant oligomers (figure 1*a*). These oligomers are largely globular, with diameters ranging from 4 to 15 nm (figure 1*b*). Most of the oligomers have diameters of 6–11 nm. Oligomers of similar sizes have also been previously reported for both Aβ40 and Aβ42 [13,34,35].

It was a surprise for us to see abundant oligomers in the Aβ42 sample. The time it took from solubilizing Aβ42 in the denaturing buffer to the preparation of TEM grids was approximately 1 h.

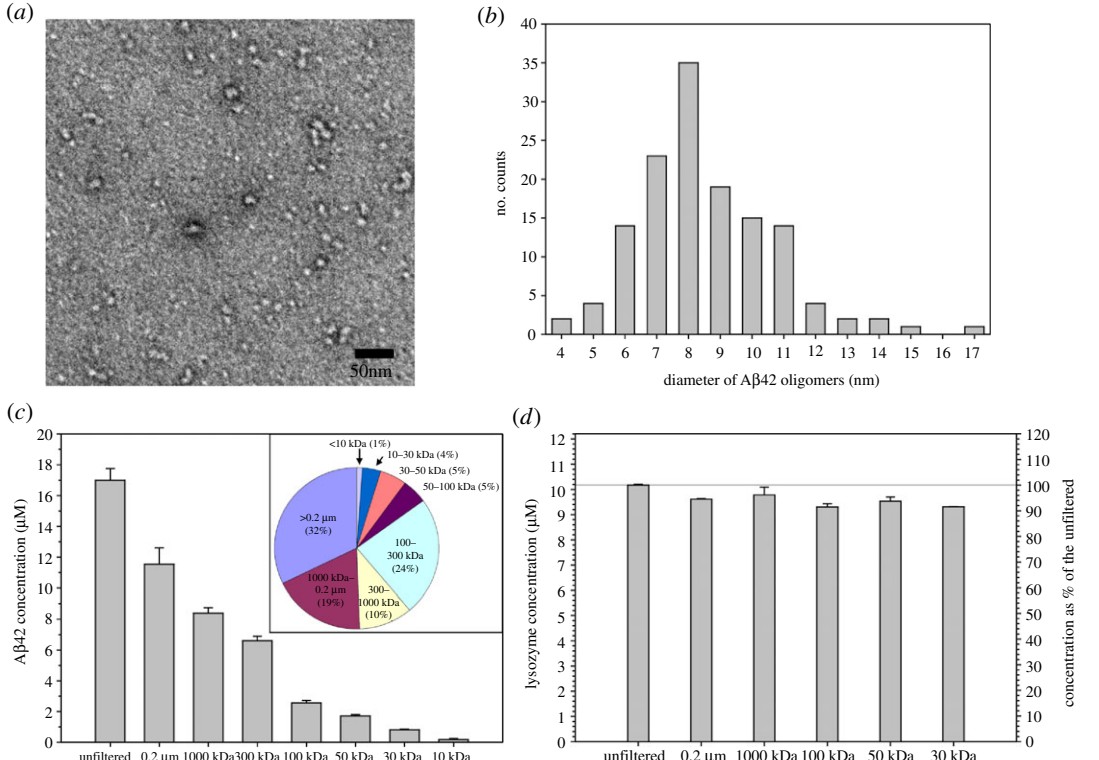

**Figure 1.** Aβ42 spontaneously forms globular oligomers of a wide range of sizes. (*a*) Transmission electron micrograph of Aβ42 samples after solubilization in a high-pH denaturing buffer and buffer exchange to PBS. (*b*) Distribution of the diameter size of Aβ42 oligomers on the electron micrograph in panel *a*. (*c*) Concentrations of Aβ42 sample upon filtration through ultrafiltration filters of various pore sizes. Error bars are standard deviations of three independent concentration measurements. Inset: percentages of different ranges of oligomer sizes. (*d*) Concentrations of hen egg white lysozyme upon filtration through ultrafiltration filters of various pore sizes. The filtration experiments were repeated three times and the error bars are standard deviations of the triplicate data. Note that different filtrates show similar concentrations for the homogeneous monomeric lysozyme.

The Aβ42 sample was kept on ice with the exception of the buffer exchange step, which took approximately 10 min. And the Aβ42 concentration was approximately 50 μM in the high-pH denaturing buffer before the buffer exchange step and was approximately 20 μM after buffer exchange to PBS. As a reference to the larger context of Aβ42 oligomer preparation, preparation of Aβ-derived diffusible ligands (ADDLs) requires incubation of 100 μM Aβ42 at 4°C for 24 h [36]. Aβ42 globulomers requires incubation of 100 μM Aβ42 at 37°C for 24 h [37]. Therefore, we expected that the Aβ42 sample consists of predominantly monomers. However, this unexpected outcome provides an opportunity to study the oligomer distribution and fibril formation starting from these oligomers. Previously, fast oligomerization has also been observed for both Aβ40 and Aβ42 [9,13,34,35,38,39], suggesting that oligomerization may be the first step of Aβ aggregation.

Next we studied the size distribution of Aβ42 oligomers. Our approach was to take advantage of the ultrafiltration filters that are available in a wide range of pore sizes. We took aliquots of the Aβ42 sample immediately after the buffer exchange step and filtered through ultrafiltration filters with seven molecular weight cutoffs: 10, 30, 50, 100, 300, 1000 kDa and 0.2 μm. Then we measured the concentration of the different filtrate samples using the fluorescamine method [40]. If all Aβ42 proteins are monomers, which are 4.5 kDa in size, then the filtrate concentration from various filters would be very similar. Differences in filtrate concentration, on the other hand, would reflect the distribution of Aβ42 oligomer size. Figure 1*c* shows the Aβ42 concentrations after filtering through various ultrafiltration filters. The filtrate concentrations decrease with decreasing pore sizes of the filters, consistent with the existence of large oligomers of various sizes. If we assume that most of the oligomers larger than the nominal size of the filter will stay in the retentate, and most of the oligomers smaller than the nominal size of the filter will stay in the filtrate, then the concentration difference between two filtrates would reflect the amount of oligomers whose sizes are in between the two filter sizes. For example, the

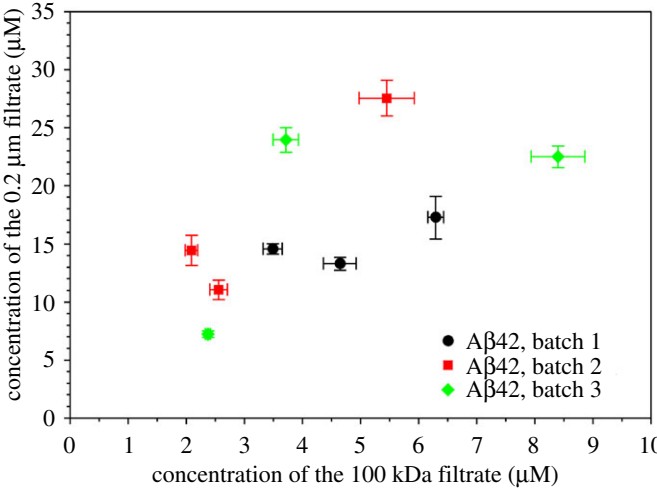

**Figure 2.** Batch variations of Aβ42 oligomer size distributions. Aβ42 samples of three batches, three samples from each batch, were filtered through 100 kDa and 0.2 μm filters, and concentrations were measured. Error bars are the standard deviations of three independent concentration measurements.

concentration of the unfiltered sample in figure 1*c* is 17.0 μM, and the concentration of the 0.2 μm filtrate is 11.5 μM, therefore, the concentration of the existing Aβ42 aggregates that are larger than 0.2 μm is the concentration difference, 5.5 μM, or 32% of the total Aβ42 sample before filtration. With the same calculation, we obtained the oligomer distribution of various size groups, and this is shown as a pie chart in figure 1*c*. Although this pie chart represents a rough estimate of oligomer size distributions, it still gives us an idea of the relative population of different oligomers. We were surprised to see that the concentration of the 10 kDa filtrate is only 1% of the unfiltered sample. Even the concentration of the small oligomers in 100 kDa filtrate is only approximately 15% of the total Aβ.

As a control, we filtered hen egg white lysozyme, a monomeric protein of 14 kDa, through 30, 50, 100, 1000 kDa, and 0.2 μm filters. The results for lysozyme show that the concentrations of the filtrate are greater than 90% of the unfiltered sample for all the filters (figure 1*d*). The 50 kDa, 1000 kDa and 0.2 μm filtrates are 94–96% of the unfiltered sample, and the 30 kDa and 100 kDa filtrates are 91–92% of the unfiltered sample. The filtration studies of lysozyme provide support for the validity of the filtration method to study oligomer size distribution.

We realize that studying oligomer size distribution with ultrafiltration filters seems to be a crude method at first glance. But we argue that ultrafiltration is actually an excellent approach for the following reasons. First, although each individual filter is not capable of resolving different sizes of oligomers, combination of filters with different molecular weight cutoffs can provide a picture that is not available through other means. Second, the time required for ultrafiltration is only 10–20 min and the filtration step can be performed at low temperatures using refrigerated centrifuge. This will help limit further oligomerization and fibrillization. One could argue that size exclusion chromatography may give better separation of oligomers of different sizes. In practice, we have not seen any reports that show discreet size exclusion peaks corresponding to Aβ oligomers of different sizes. There are reports in which researchers have used fraction numbers as an indication of oligomer size. When used this way, size exclusion chromatography lacks the resolution required to distinguish Aβ oligomers of different sizes. Ultrafiltration filters have been used by other groups as a way to separate Aβ oligomers into different size groups, and have successfully shown that oligomers separated using ultrafiltration filters have different bioactivities [41,42].

We next investigated the sample-to-sample and batch-to-batch variations in terms of the size distribution of Aβ42 oligomers. We studied three purification batches of Aβ42. Each batch of Aβ42 was prepared from independently cultured *E. coli* cells, and multiple tubes of Aβ42 were obtained from each batch of preparation. Within each batch, we investigated three tubes of Aβ42 sample. In total, nine tubes of HFIP-treated Aβ42 powder were separately dissolved in the high-pH denaturing buffer called CG. The concentration of Aβ42 in CG buffer was 50 μM for all nine tubes. After buffer exchange to PBS buffer, two aliquots of 400 μl from each sample were filtered through 0.2 μm or 100 kDa filters, and the concentration of the filtrate was measured using the fluorescamine method [40]. The results of measured concentrations are shown in figure 2. The concentrations of both the 0.2 μm and 100 kDa filtrates show a wide range even though the starting concentrations in CG buffer

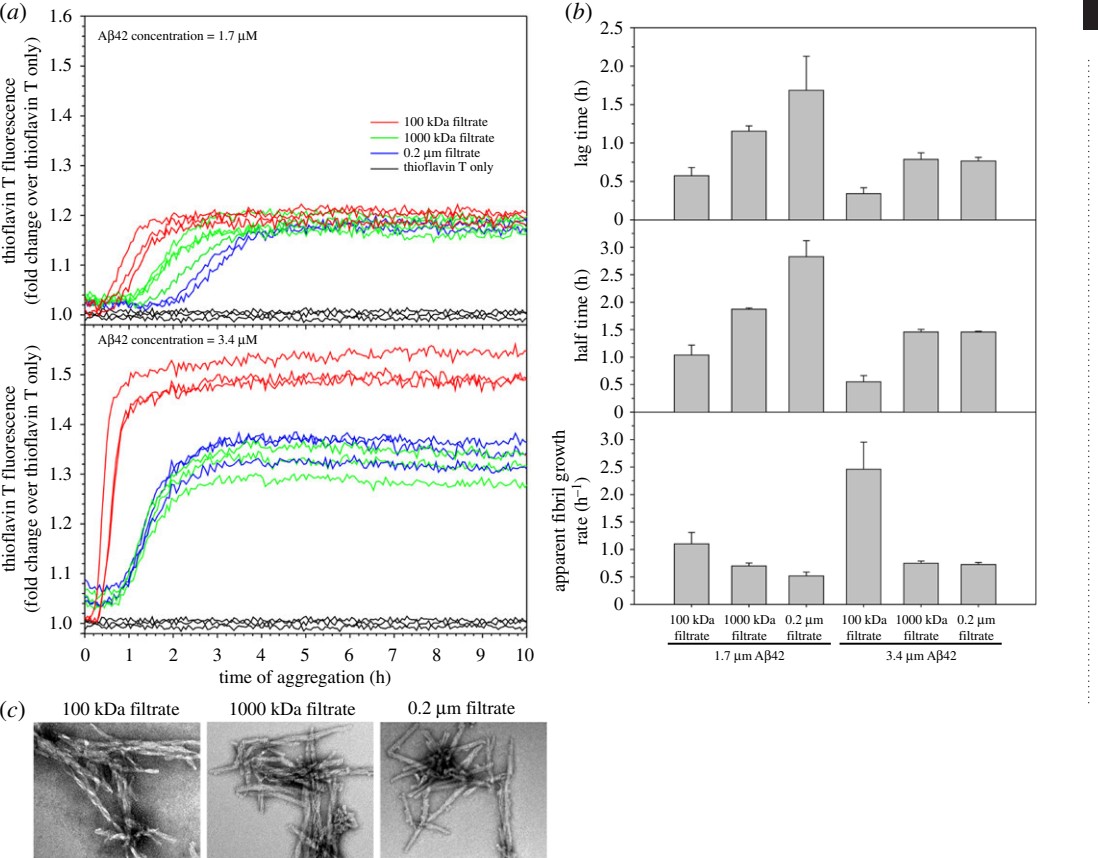

**Figure 3.** Smaller Aβ42 oligomers form fibrils faster than larger oligomers. (*a*) Fibrillization kinetics for Aβ42 samples upon filtration using three ultrafiltration filters (100 kDa, 1000 kDa, and 0.2 μm). The aggregation experiments were set up at two Aβ42 concentrations: 1.7 μM and 3.4 μM. Three repeats were performed for each filtrate and concentration. Aggregation was monitored with thioflavin T fluorescence and the same thioflavin T concentration (20 μM) was used for all the aggregations. The fluorescence data were presented using fold change, which was calculated by dividing the fluorescence reading by the average of the thioflavin T only samples at each time point of measurement. (*b*) Kinetic parameters, lag time, half time and fibril growth rate, were extracted from the aggregation data by fitting to a sigmoidal function (see Material and methods). (*c*) Morphology of aggregated Aβ42 samples as described in panel *a*. The samples were taken at 24-h time point of aggregation and examined with TEM. Note that Aβ42 oligomers of different sizes all lead to the formation of fibrils.

are all the same at 50 μM. These results suggest that the initial oligomerization process is controlled by a stochastic factor. As a result, the oligomer size distribution is not reproducible from one tube of sample to another. This stochastic control of oligomerization may be similar in principle to the stochastic nucleation for Aβ aggregation proposed by Fändrich and co-workers [43].

## 2.2. Aβ42 samples enriched with smaller oligomers form fibrils faster than Aβ42 samples enriched with larger oligomers

We then investigated how these Aβ42 samples form fibrils after ultrafiltration. We studied three filtrate samples using the 100 kDa, 1000 kDa and 0.2 μm filters. These Aβ42 filtrates all started from a single tube of HFIP-treated Aβ42, which was dissolved in the high-pH denaturing buffer, buffer exchanged to PBS, and then split into three aliquots for ultrafiltration with 100 kDa, 1000 kDa and 0.2 μm filters. The 100 kDa filtrate represents Aβ42 monomers and small oligomers; 0.2 μm filtrate has a majority of larger oligomers; and the 1000 kDa filtrate represents a sample that is devoid of largest oligomers. After concentration measurements using fluorescamine, we set up fibrillization experiments at 37°C without agitation. Two Aβ42 concentrations at 1.7 and 3.4 μM, and three repeats for each concentration, were incubated in a plate reader to allow continuous monitoring of aggregation. The aggregation curves are shown in figure 3*a*. The aggregation data were fitted to a sigmoidal equation to obtain the lag time, half time and apparent fibril growth rate (see Material and methods and

electronic supplementary material, figure S1). The 100 kDa filtrate samples form fibrils faster than both the 1000 kDa and 0.2 μm filtrates, at both 1.7 μM and 3.4 μM concentrations. At lower concentration (1.7 μM), the 1000 kDa filtrate forms fibrils faster than the 0.2 μm filtrate, but the difference between the 1000 kDa and 0.2 μm filtrates disappeared at higher concentration (3.4 μM), suggesting that faster aggregation may obscure differences in aggregation kinetics between some samples.

We also noticed that the 100 kDa filtrate has faster apparent fibril growth rate than both the 1000 kDa and 0.2 μm filtrates (figure 3b). Furthermore, the fibril growth rate for the 100 kDa filtrate more than doubles from 1.7 to 3.4 μM Aβ42. By contrast, the 1000 kDa and 0.2 μm filtrates show marginal increase when Aβ42 concentration is increased from 1.7 to 3.4 μM. Two main factors contribute to the apparent fibril growth rate: the rate of fibril elongation and the concentration of fibril nuclei. Because the same Aβ42 protein was used, the rate of fibril elongation is likely similar for different filtrates. Therefore, we speculate that the high fibril growth rate for the 100 kDa sample is mainly due to higher concentrations of fibril nuclei than the 1000 kDa and 0.2 μm filtrates.

We examined the aggregation products after 24 h of incubation using TEM. The electron micrographs show fibrillar morphology for all three filtrate samples (figure 3c), confirming that the aggregation reactions we are studying here are fibrillization.

We considered the possibility that the differences in aggregation kinetics resulted from concentration differences in the aggregation reaction. In other words, we may underestimate the concentration of the 100 kDa filtrate, or overestimate the concentration of the 1000 kDa and 0.2 μm filtrates. We do not believe that this is likely for the following reasons. First, we used fluorescamine to measure the concentration of the Aβ filtrate [40]. Fluorescamine is a small molecule of 276 Da. It reacts with the primary amine groups of N-terminus or lysine. In oligomers, it is possible that the primary amines are buried inside the protein core and are thus not accessible to fluorescamine. As a result, this would lead to an underestimate of Aβ concentration. If this were the case, the concentration of larger oligomers would be underestimated compared to smaller oligomers. The aggregation kinetics show that samples with larger oligomers form fibrils slower, in contrast to this possibility. Second, the fluorescence intensity at aggregation plateau suggests that our concentration measurement is accurate. Xue et al. [44] showed that thioflavin T fluorescence is a quantitative measure of fibril quantity. For Aβ42 aggregation at 1.7 μM, all three filtrates reach similar fluorescence intensity at aggregation plateau. For Aβ42 aggregation at 3.4 μM, the 100 kDa filtrate reached a higher fluorescence intensity than the 1000 kDa and 0.2 μm filtrates. We speculate that the 100 kDa filtrate forms fibrils of a different structure from the 1000 kDa and 0.2 μm filtrates, and this point is further discussed below. Third, even if we compare the 100 kDa filtrate at 1.7 μM with the 1000 kDa and 0.2 μm filtrates at 3.4 μM, the 100 kDa filtrate still forms fibrils faster, and has a faster apparent fibril growth rate, suggesting that the difference in fibrillization kinetics between smaller and larger oligomers is not due to inaccurate concentration measurements.

We next investigated the reproducibility of our observation with a different Aβ preparation batch (figure 4a,b), and an Aβ sample without HFIP treatment (figure 4c,d). In both cases, we observed that the 100 kDa filtrate forms fibrils faster than the 0.2 μm filtrate, and the 1000 kDa filtrate forms fibrils at a rate that is in between the 100 kDa and 0.2 μm filtrates. Similarly, we also observed that the fibril growth rate of the 100 kDa filtrate increases dramatically from 1.7 to 3.4 μM Aβ42, while the 1000 kDa and 0.2 μm filtrates show marginally higher fibril growth rate at higher Aβ42 concentrations.

In addition to the faster fibrillization rate, smaller Aβ42 oligomers often show higher thioflavin T fluorescence at aggregation plateau, especially at the higher concentration that we studied (figures 3 and 4). Xue et al. [44] previously showed that thioflavin T fluorescence is a quantitative measure of fibril amount. Since we are comparing fibril formation at the same Aβ42 concentration, the difference in thioflavin T fluorescence suggests a possibility that the underlying fibril structure, not just the amount of fibrils, is different for smaller and larger oligomers. Although the best evidence may come from the structures of fibrils formed by the smaller and larger oligomers, structural determination of these fibrils is a challenging task in itself and is beyond the scope this work. However, there are other ways to study the structure based on the relationship between protein structure and its thermodynamic stability. In a separate study of the Guo laboratory (unpublished data, GP & ZG, 2019), we have found that epigallocatechin gallate (EGCG), a main component of green tea catechins, is capable of solubilizing Aβ fibrils, and this effect depends on how the fibrils were prepared. Therefore, we investigated how EGCG affects the fibrillization of the 100 kDa and 0.2 μm filtrates. Figure 5 shows the aggregation of 5 μM Aβ42, filtered through 100 kDa or 0.2 μm filters, in the presence of 25 μM EGCG. For the 0.2 μm filtrate, EGCG leads to a steady decrease in thioflavin T fluorescence after the fibril growth phase (figure 5a). By contrast, the thioflavin T fluorescence for the 100 kDa filtrate remained stable after the aggregation has reached plateau (figure 5b). These results suggest a difference in stability for the fibrils formed by the

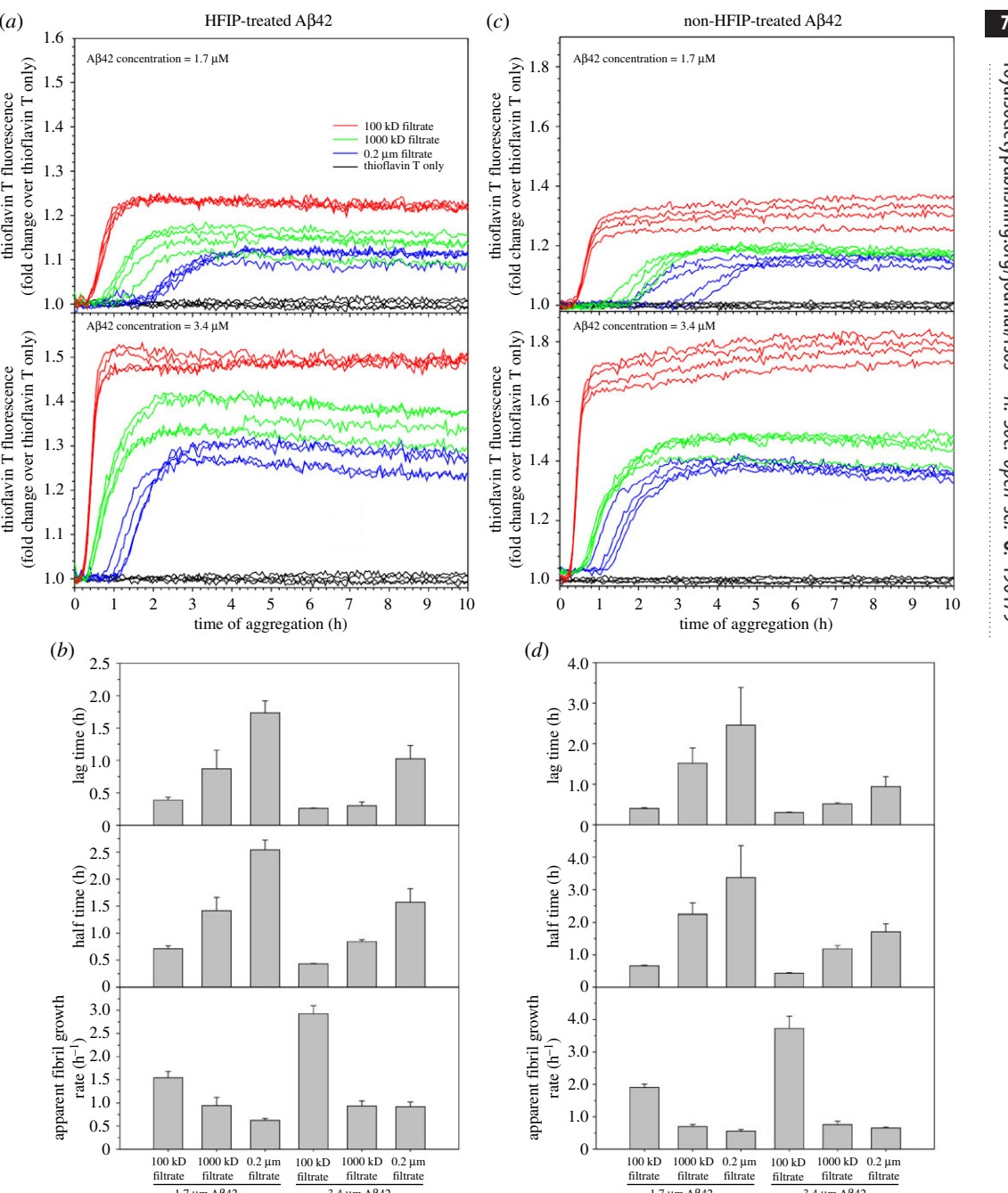

**Figure 4.** Batch variations and the effect of HFIP treatment on the fibrillization of different Aβ42 filtrates. (*a,b*) Aβ42 sample of a different batch from the one in figure 3 was used for the ultrafiltration and aggregation. Four repeats of each filtrate and concentration were studied. The aggregation curves are shown in *a*, and kinetic parameters from fitting to a sigmoidal function are shown in *b*. (*c,d*) Aβ42 sample of the same batch as the one in figure 3, but without HFIP treatment, was used for ultrafiltration and aggregation. Four repeats of each filtrate and concentration were studied. The aggregation curves are shown in *c*, and kinetic parameters from fitting to a sigmoidal function are shown in *d*.

100 kDa and 0.2 μm filtrates. While the 0.2 μm fibrils can be solubilized, although not completely, by EGCG, the 100 kDa fibrils are resistant to EGCG solubilization at this combination of Aβ and EGCG concentrations. We attribute this difference to the underlying structure of the 100 kDa and 0.2 μm fibrils.

The fibrillization kinetics of different Aβ42 oligomers have implications regarding the mechanism of Aβ aggregation. Here we consider three potential pathways of Aβ aggregation that involves Aβ oligomers. In the first pathway (figure 6*a*), Aβ aggregates to form oligomers, and then oligomers grow in size and eventually form fibrils. Our results are not in favour of this pathway because larger oligomers form fibrils at a slower rate (figures 3 and 4). In the second pathway (figure 6*b*), Aβ

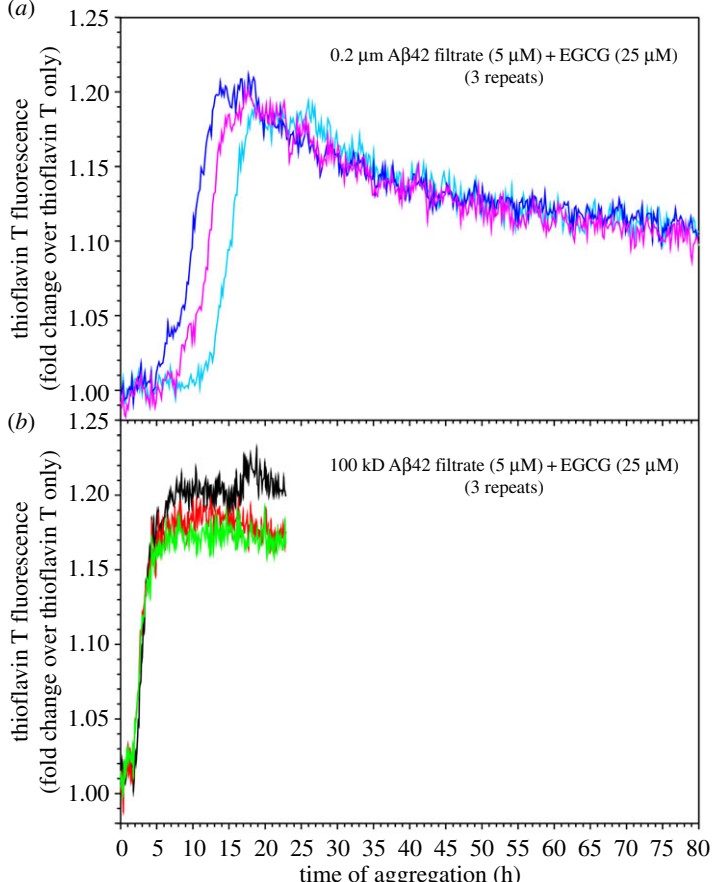

**Figure 5.** Aβ42 fibrillization in the presence of EGCG. Aggregation of Aβ42 samples (5 μM) filtered through 0.2 μm (a) and 100 kDa (b) filters was set up in the presence of EGCG (25 μM). Note that the 0.2 μm filtrate sample shows a gradual decrease in fluorescence after the fibril growth phase, which is attributed to the solubilization of Aβ42 fibrils by EGCG. By contrast, the fluorescence signal for the 100 kDa filtrate remains stable after the aggregation reaches plateau.

monomers and oligomers are in fast equilibrium, and different Aβ species form fibrils through a common step, such as nucleation from monomers. This pathway predicts that different filtrates would form fibrils with similar fibril growth rate and similar final thioflavin T fluorescence, with the main difference being lag time. Our results are not consistent with this model (figures 3 and 4). In the third pathway (figure 6c), Aβ aggregates to form oligomers of different sizes, and different oligomers proceed to form fibrils without interconversion between different oligomers. This would explain the different thioflavin T fluorescence amplitude from different oligomers upon fibrillization, and the potential structural difference between these fibrils (figure 5). The step of oligomer to fibril formation is via nucleated conformation conversion, which has been proposed previously for Aβ [9,10]. Similar mechanisms have also been proposed for other amyloid systems [45,46]. The contribution of our model here is to include different types of oligomers and to suggest that different oligomers form fibrils in parallel fibrillization pathways. Therefore, our model suggests that Aβ42 samples of smaller oligomers would form a higher concentration of fibril nuclei, consistent with the higher apparent fibril growth rate (figures 3 and 4). The shorter lag time for the fibrillization of smaller oligomers is likely due to a faster conversion rate from oligomers to fibril nuclei. Previously, we have also suggested that the mechanism of this nucleated conformational conversion is a β-strand rotation, which converts antiparallel β-structure in oligomers to parallel β-sheet structure in fibril nuclei [30]. A similar structural conversion was also proposed by Fu et al. [10] based on NMR studies.

Our model of Aβ aggregation suggests a link between oligomer heterogeneity and fibril polymorphism. The issue of fibril polymorphism has been long recognized in the amyloid research community [47]. Different fibril polymorphs have been observed even in the same sample prepared either *in vitro* [48] or *in vivo* [49], and may lead to different pathology in Alzheimer's disease. Different underlying structures of Aβ fibrils have been found in different clinical subtypes of Alzheimer's disease patients [50–52]. Our results suggest that Aβ42 oligomers of different sizes may

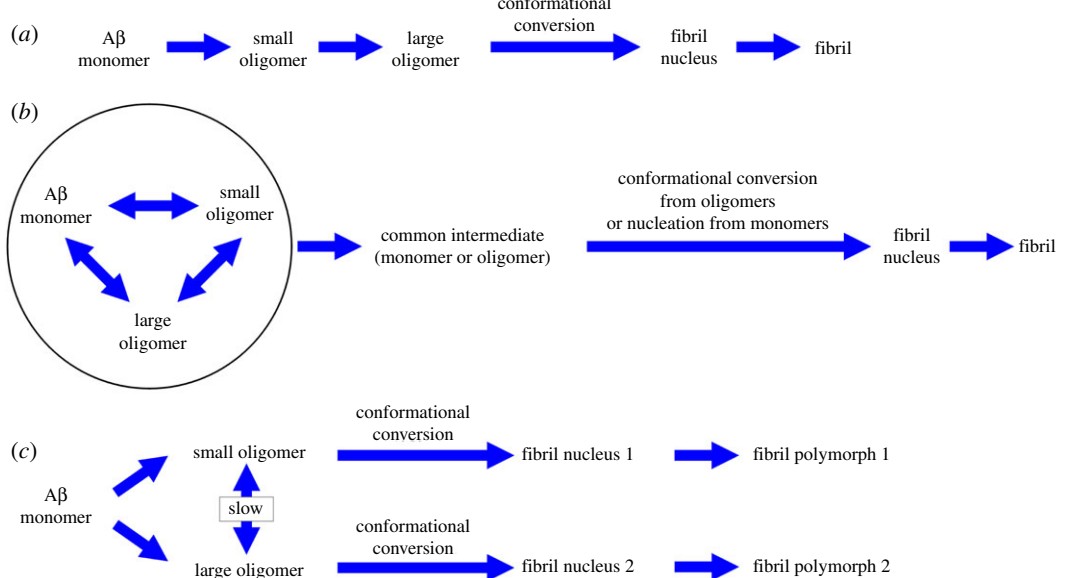

**Figure 6.** Implications on potential fibrillization pathways of Aβ. (*a*) Aβ first forms oligomers, which grow in size and eventually convert to fibrils. This model predicts that larger oligomers form fibrils faster than smaller oligomers. (*b*) Aβ monomers and different oligomers are in fast equilibrium and different Aβ species form fibrils through a common intermediate. This model predicts that Aβ from different starting states may form fibrils with different length of lag phase, but should have similar characteristics for the fibril growth phase (such as fibril growth rate and fluorescence at aggregation plateau). (*c*) Aβ aggregates to form different oligomers, and the interconversion between oligomers is slow compared to fibrillization. Different oligomers form fibrils independently, and could form fibrils of different structures.

form fibrils of different structures. Therefore, oligomer heterogeneity leads to fibril polymorphism. The heterogeneity of oligomers is likely a result of the stochastic nature of the oligomerization process. Due to the presence of two hydrophobic regions (the central hydrophobic cluster and the C-terminal region) in Aβ sequence and their importance in Aβ aggregation, the initial oligomerization process is likely driven by hydrophobic collapse. Similar to protein folding where the transition state is similar to the native state of a folded protein [53], the structures of the oligomers also dictate the final structures of fibrils that they eventually form. Given the diversity of Aβ42 oligomers that have been identified *in vitro* and *in vivo*, and the highly polymorphic nature of Aβ42 fibrils, our results here point to the potential convergence of these two lines of research inquires.

# 3. Material and methods

## 3.1. Preparation of Aβ42 proteins

Detailed protocols for Aβ42 preparation have been described previously [25,44]. Briefly, Aβ protein was expressed in *Escherichia coli* as a fusion protein, GroES-ubiquitin-Aβ, and was purified using a nickel column in a high-pH denaturing buffer. After purification, the fusion protein was cleaved to obtain full-length Aβ. Full-length Aβ was then buffer exchanged to 30 mM ammonium acetate (pH 10). At this stage, Aβ42 proteins in ammonium acetate from several purifications were pooled together, and then were aliquoted to small volumes in 1.5-ml microcentrifuge tubes for lyophilization. This pooled Aβ42 sample is called a purification batch. Each batch of Aβ consists of multiple tubes of Aβ powder, and each tube of Aβ is identical to other tubes within the same batch. The lyophilized Aβ powder was then treated with HFIP by incubating at approximately 100 μM Aβ concentration at room temperature with shaking (1000 r.p.m.) for 24 h. The HFIP was allowed to dry overnight in a chemical hood. The resulting Aβ film is stored at −80°C.

## 3.2. Ultrafiltration to prepare various Aβ42 filtrate samples

HFIP-treated Aβ42 was dissolved in a high-pH denaturing buffer called CG (20 mM CAPS, 7 M GdnHCl, pH 11) to a final Aβ concentration of 30−50 μM. Then the Aβ sample was buffer exchanged to PBS using

a 5-ml HiTrap desalting column (GE Healthcare). Depending on the requirement of Aβ volume, multiple tubes of Aβ may be prepared and pooled together after buffer exchange step. Following buffer exchange, 400 μl of Aβ was pipetted to ultrafiltration filters of different molecular weight cutoff sizes ranging from 10 kDa to 0.2 μm and centrifuged at 4°C at 14 000$g$. For all the experiments described here, the ultrafiltration filters of 10, 30, 50, 100, 300 and 1000 kDa were obtained from Sartorious (Vivaspin 500 product line), and the 0.2 μm filters were obtained from PALL (Nanosep product line). All filters were pre-washed with PBS buffer. The centrifugation was monitored so that most of the sample is in the filtrate and the retentate volume is minimal.

## 3.3. Measurement of Aβ concentration using the fluorescamine method

Aβ concentration was measured using the fluorescamine method as previously described [40]. A standard curve using hen egg white lysozyme (Fisher Bioreagents, crystalline powder) was prepared for each experiment of concentration measurement. The range of amine concentrations for the standard curve is 0.3–7 μM. The volume of each assay was 50 μl, containing 500 μM fluorescamine. The assay for Aβ was performed the same way as the lysozyme standard. The Aβ sample was also diluted when needed so that the fluorescence reading is in the middle of the concentration range of the standard curve. The concentration of each Aβ sample was measured in triplicates.

## 3.4. Ultrafiltration of hen egg white lysozyme

As a control experiment, we prepared a lysozyme solution at 10.17 μM. Then 400 μl of aliquots were filtered through filters of five molecular weight cutoff sizes: 30 kDa, 50 kDa, 100 kDa, 1000 kDa and 0.2 μm. These filters were obtained from the same manufacturers as described above for Aβ ultrafiltration experiments. Following centrifugation, the lysozyme concentration of each filtrate was measured using absorbance at 280 nm and an extinction coefficient of 38 mM$^{-1}$ cm$^{-1}$ [54]. This ultrafiltration experiment was repeated for two additional times to obtain final concentration data in triplicates.

## 3.5. Transmission electron microscopy

Aβ samples were applied on to glow-discharged copper grids (400 mesh formvar/carbon film, Ted Pella) and stained with 2% uranyl acetate. The EM grids were then examined using a FEI T12 electron microscope with an accelerating voltage of 120 kV.

## 3.6. Fibrillization of Aβ42 samples following ultrafiltration

For each of the three aggregation experiments reported in figures 3 and 4$a$,$b$, one tube of Aβ42 (HFIP-treated for figures 3 and 4$a$, non-HFIP-treated for figure 4$b$) was dissolved in 1 ml of CG buffer, and then buffer exchanged to PBS. Following buffer exchange, three aliquots of 400 μl Aβ42 were filtered through the 0.2 μm, 1000 kDa and 100 kDa ultrafiltration filters at the same time using a refrigerated centrifuge. Centrifugation was monitored so that most of the volume is in the filtrate and the retentate volume is minimal. Then the concentration of the filtrates was determined using the fluorescamine method as described above. Aβ42 fibrillization assays were set up at two Aβ42 concentrations: 1.7 and 3.4 μM. A 50-μl assay was prepared by mixing 5 μl of thioflavin T stock (200 μM in PBS, freshly prepared), Aβ42 filtrate samples and PBS buffer. The thioflavin T control was prepared using 5 μl of thioflavin T stock and 45 μl of PBS buffer. The volumes of the Aβ42 filtrate and PBS were calculated based on the final Aβ42 concentration. Each aggregation condition (different filters and different Aβ42 concentrations) was performed either in triplicates or quadruplicates as indicated in the figure legends. All the mixing steps were performed on ice. Finally, the fibrillization mixtures (50 μl each) were transferred to a black 384-well Non-binding Surface microplate with clear bottom (Corning product 3655) and sealed with a polyester-based sealing film (Corning product PCR-SP). The fibrillization was started by incubation at 37°C in a Victor 3 V plate reader (Perkin Elmer) without agitation. The thioflavin T fluorescence was measured from the bottom of the plate with an excitation filter of 450 nm and an emission filter of 490 nm. The aggregation data are reported as fold change in fluorescence by dividing the average of thioflavin T fluorescence at each time point of measurements.

## 3.7. Fibrillization of Aβ42 samples in the presence of EGCG

One tube of HFIP-treated Aβ42 was dissolved in CG buffer and buffer exchanged to PBS. Then aliquots of 400 µl samples were filtered through the 0.2 µm and 100 kDa ultrafiltration filters. The concentration of Aβ filtrates was determined using the fluorescamine method described above. EGCG was obtained from Tocris (greater than 98% purity, catalogue number 4524) and was dissolved in deionized water. The concentration of EGCG was calculated based on the amount (weight) and water volume. The final aggregation was set up to have 5 µM of Aβ42 (either 0.2 µm filtrate or 100 kDa filtrate), 25 µM EGCG and 20 µM thioflavin T. All the mixing steps were performed on ice, and the aggregation was monitored using a plate reader as described above.

## 3.8. Fitting of fibrillization data

To obtain the kinetic parameters, the fibrillization data were fitted to the equation below, which was based on the sigmoidal equation described by Nielsen *et al.* [55] with modifications.

$$S = S_I + k_I \cdot t + \frac{(S_F + k_F \cdot t) - (S_I + k_I \cdot t)}{1 + e^{-((t - t_{50})/\tau)}}, \tag{3.1}$$

where $S$ is the fluorescence intensity, $S_I$ and $k_I$ are the fluorescence and slope in the lag phase, $S_F$ and $k_F$ are the fluorescence and slope in the plateau phase, $t$ is the time of aggregation, $t_{50}$ is the time to reach 50% of maximal fluorescence, and $\tau$ is the time it takes for the fluorescence signal to change from 26.9 to 50%, or from 50 to 73.1%, of the maximum fluorescence. The lag time of aggregation is given by $t_{50} - 2\tau$, and the maximum fibril growth rate is given by $1/4\tau$. The fits are shown in electronic supplementary material, figures S1–S3.

Data accessibility. Data available from the Dryad Digital Repository: https://doi.org/10.5061/dryad.7n4c218 [56].
Authors' contributions. C.X. and J.T. designed and carried out experiments, analysed data and drafted the manuscript. H.W., G.P. and F.H. carried out experiments, analysed data and revised the manuscript. Z.G. conceived and supervised the study, designed the experiments and drafted the manuscript. All authors gave final approval for publication.
Competing interests. The authors declare no competing interests.
Funding. This work was supported by the National Institutes of Health (grant no. R01AG050687).

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
