## [Reviewer comments · Royal Society Open Science]

Review History

RSOS-190179.R0 (Original submission)

Review form: Reviewer 1

Is the manuscript scientifically sound in its present form?

Yes

Are the interpretations and conclusions justified by the results?

Yes

Is the language acceptable?

Yes

Is it clear how to access all supporting data?

Yes

Do you have any ethical concerns with this paper?

No

Have you any concerns about statistical analyses in this paper?

No

Recommendation?

Accept as is

Comments to the Author(s)

In this work, the authors studied the oligomer heterogeneity of A β 42 and the kinetics of oligomer and fibril formation. This work is important since A β is a well-known amyloid protein but its oligomerization and fibrillization are still not fully understood. Here, the authors used a combination of practical techniques, including ultrafiltration, TEM, and fluorescence methods, and observed a wide range of oligomer sizes with distinct fibril formation kinetics. The study was well executed and the data support the conclusions.

Minor comments:

- 1). Why do the smaller oligomers form fibrils faster than the larger ones? Perhaps provide some rationale or speculation in the discussion.
- 2). Page 7, line 4, could the authors provide a reference for "a separate study of the Guo laboratory"?
- 3). Figure 1, would the size distribution of the oligomers be different with longer incubation times than 10-20 minutes after the buffer exchange?
- 4). Figure 2, what were the concentrations of the original unfiltered samples?

Review form: Reviewer 2

Is the manuscript scientifically sound in its present form?

Yes

Are the interpretations and conclusions justified by the results?

Yes

Is the language acceptable?

Yes

Is it clear how to access all supporting data?

Yes

Do you have any ethical concerns with this paper?

No

Have you any concerns about statistical analyses in this paper?

No

Recommendation?

Major revision is needed (please make suggestions in comments)

Comments to the Author(s)

The relationship between oligomers and amyloid fibrils in the aggregation process has been clarified in these papers (PMID: 27500912; 27226198; 28671122; 29338255; 30480261). The authors should add this information in the introduction.

Moreover, using electron microscopy and others methods it has been shown for Abeta peptides and their fragments and also for insulin and the peptide fragments from the BGL2P protein that amyloid fibrils consist of oligomers (PMID: 27500912; 27226198; 28671122; 29338255; 30480261; 30686252).

The peak of sizes on Fig. 1b (8 nm) corresponds to the size of oligomers that contribute to the formation of the amyloid fibrils for Abeta40 and Abeta42 (see these papers PMID: 28671122; 27016282). According to the results of this manuscript the amyloid fibrils also are constructed from the oligomers.

It should be underline that folding nucleus for globular proteins and for formation of fibrils are different (PMID: 24404849), because as has been demonstrated the primary folding nucleus for Abeta40 consists of 2 monomers and for Abeta42 - 3 monomers (see the papers PMID: 27016282; 28671122), but folding nucleus for Abeta peptide is different (for comparison please see this paper PMID: 15632297).

The authors should accent some points in the paper taking into account this new information.

Review form: Reviewer 3

Is the manuscript scientifically sound in its present form?

Yes

Are the interpretations and conclusions justified by the results?

Yes

Is the language acceptable?

Yes

Is it clear how to access all supporting data?

Yes

Do you have any ethical concerns with this paper?

No

Have you any concerns about statistical analyses in this paper?

No

Recommendation?

Accept with minor revision (please list in comments)

Comments to the Author(s)

Aggregation of Abeta peptide has been considered to be closely related with the pathology of Alzheimer's disease, however the stochastic characteristic of oligomer and fibril formation of Abeta makes it particularly challenging for reliable investigation of the fundamental mechanistic roles of the peptide in AD pathology. Xue et al. in this manuscript reported an elegant study of the potential correlation of oligomer heterogeneity and fibril polymorphism of Abeta. They performed ultrafiltration with variant cutoff size to separate Abeta oligomers with different size. The aggregation kinetics of these oligomers were monitored and compared, showing that smaller oligomers surprisingly forms fibrils with a faster rate. The difference of EGCG solubilization of fibrils from oligomers with distinct size further indicate the structural variances of the fibrils. The experiments were carried out thoroughly to confirm the aggregation difference observed, and the data was carefully analyzed and well discussed. The results provide new mechanistic insight into

the polymorphism nature of the amyloid structures. The manuscript is suitable for publication, after considering some minor points addressed below:

1. "ADDL" was mentioned in the second paragraph of page 4. The full meaning of this abbreviation need to be provided.
2. The authors reported a surprising fast oligomer formation rate of Abeta42 sample. Indeed, the fast formation of oligomers of Abeta42/42 has been realized in literatures (e.g., fast oligomer formation within hours for Abet40, a moderate aggregation prone one compared to Abeta42, Elbassal et al., JPCC, 2017, 20007). The authors may include more relevant literature reports in the discussion.

Decision letter (RSOS-190179.R0)

05-Apr-2019

Dear Dr Guo,

The editors assigned to your paper ("A β 42 fibril formation from predominantly oligomeric samples suggests a link between oligomer heterogeneity and fibril polymorphism") have now received comments from reviewers. We would like you to revise your paper in accordance with the referee and Associate Editor suggestions which can be found below (not including confidential reports to the Editor). Please note this decision does not guarantee eventual acceptance.

Please submit a copy of your revised paper before 28-Apr-2019. Please note that the revision deadline will expire at 00.00am on this date. If we do not hear from you within this time then it will be assumed that the paper has been withdrawn. In exceptional circumstances, extensions may be possible if agreed with the Editorial Office in advance. We do not allow multiple rounds of revision so we urge you to make every effort to fully address all of the comments at this stage. If deemed necessary by the Editors, your manuscript will be sent back to one or more of the original reviewers for assessment. If the original reviewers are not available, we may invite new reviewers.

If your study uses humans or animals please include details of the ethical approval received, including the name of the committee that granted approval. For human studies please also detail

whether informed consent was obtained. For field studies on animals please include details of all permissions, licences and/or approvals granted to carry out the fieldwork.

- Data accessibility

If you wish to submit your supporting data or code to Dryad (<http://datadryad.org/>), or modify your current submission to dryad, please use the following link:
<http://datadryad.org/submit?journalID=RSOS&manu=RSOS-190179>

- Competing interests

- Authors' contributions

- Acknowledgements

- Funding statement

Kind regards,
Andrew Dunn
Royal Society Open Science Editorial Office

Editor comments:

The referees have each provided a number of comments to improve your work. Please ensure that you fully respond to and incorporate their recommendations. You will not be granted a second opportunity to revise the manuscript.

Comments to Author:

Reviewers' Comments to Author:

Reviewer: 1

Comments to the Author(s)

In this work, the authors studied the oligomer heterogeneity of A β 42 and the kinetics of oligomer and fibril formation. This work is important since A β is a well-known amyloid protein but its oligomerization and fibrillization are still not fully understood. Here, the authors used a combination of practical techniques, including ultrafiltration, TEM, and fluorescence methods, and observed a wide range of oligomer sizes with distinct fibril formation kinetics. The study was well executed and the data support the conclusions.

Minor comments:

- 1). Why do the smaller oligomers form fibrils faster than the larger ones? Perhaps provide some rationale or speculation in the discussion.
- 2). Page 7, line 4, could the authors provide a reference for "a separate study of the Guo laboratory"?
- 3). Figure 1, would the size distribution of the oligomers be different with longer incubation times than 10-20 minutes after the buffer exchange?
- 4). Figure 2, what were the concentrations of the original unfiltered samples?

Reviewer: 2

Comments to the Author(s)

The relationship between oligomers and amyloid fibrils in the aggregation process has been clarified in these papers (PMID: 27500912; 27226198; 28671122; 29338255; 30480261). The authors should add this information in the introduction.

Moreover, using electron microscopy and others methods it has been shown for Abeta peptides and their fragments and also for insulin and the peptide fragments from the BGL2P protein that amyloid fibrils consist of oligomers (PMID: 27500912; 27226198; 28671122; 29338255; 30480261; 30686252).

The peak of sizes on Fig. 1b (8 nm) corresponds to the size of oligomers that contribute to the formation of the amyloid fibrils for Abeta40 and Abeta42 (see these papers PMID: 28671122; 27016282). According to the results of this manuscript the amyloid fibrils also are constructed from the oligomers.

It should be underline that folding nucleus for globular proteins and for formation of fibrils are different (PMID: 24404849), because as has been demonstrated the primary folding nucleus for Abeta40 consists of 2 monomers and for Abeta42 - 3 monomers (see the papers PMID: 27016282; 28671122), but folding nucleus for Abeta peptide is different (for comparison please see this paper PMID: 15632297).

The authors should accent some points in the paper taking into account this new information.

Reviewer: 3

Comments to the Author(s)

Aggregation of Abeta peptide has been considered to be closely related with the pathology of Alzheimer's disease, however the stochastic characteristic of oligomer and fibril formation of Abeta makes it particularly challenging for reliable investigation of the fundamental mechanistic roles of the peptide in AD pathology. Xue et al. in this manuscript reported an elegant study of the potential correlation of oligomer heterogeneity and fibril polymorphism of Abeta. They performed ultrafiltration with variant cutoff size to separate Abeta oligomers with different size. The aggregation kinetics of these oligomers were monitored and compared, showing that smaller oligomers surprisingly forms fibrils with a faster rate. The difference of EGCG solubilization of fibrils from oligomers with distinct size further indicate the structural variances of the fibrils. The experiments were carried out thoroughly to confirm the aggregation difference observed, and the data was carefully analyzed and well discussed. The results provide new mechanistic insight into the polymorphism nature of the amyloid structures. The manuscript is suitable for publication, after considering some minor points addressed below:

1. "ADDL" was mentioned in the second paragraph of page 4. The full meaning of this abbreviation need to be provided.
2. The authors reported a surprising fast oligomer formation rate of Abeta42 sample. Indeed, the fast formation of oligomers of Abeta42/42 has been realized in literatures (e.g., fast oligomer formation within hours for Abeta40, a moderate aggregation prone one compared to Abeta42, Elbassal et al., JPCC, 2017, 20007). The authors may include more relevant literature reports in the discussion.

Author's Response to Decision Letter for (RSOS-190179.R0)

See Appendix A.

RSOS-190179.R1 (Revision)

Review form: Reviewer 1

Is the manuscript scientifically sound in its present form?

Yes

Are the interpretations and conclusions justified by the results?

Yes

Is the language acceptable?

Yes

Is it clear how to access all supporting data?

Yes

Do you have any ethical concerns with this paper?

No

Have you any concerns about statistical analyses in this paper?

No

Recommendation?

Accept as is

Comments to the Author(s)

No further comments.

Review form: Reviewer 2

Is the manuscript scientifically sound in its present form?

Yes

Are the interpretations and conclusions justified by the results?

Yes

Is the language acceptable?

Yes

Is it clear how to access all supporting data?

Yes

Do you have any ethical concerns with this paper?

No

Have you any concerns about statistical analyses in this paper?

No

Recommendation?

Accept with minor revision (please list in comments)

Comments to the Author(s)

I have only one comment, this paper from the references has pages: "New Mechanism of Amyloid Fibril Formation." Galzitskaya O. Curr Protein Pept Sci. 2019;20(6):630-640.

Review form: Reviewer 3

Is the manuscript scientifically sound in its present form?

Yes

Are the interpretations and conclusions justified by the results?

Yes

Is the language acceptable?

Yes

Is it clear how to access all supporting data?

Yes

Do you have any ethical concerns with this paper?

No

Have you any concerns about statistical analyses in this paper?

No

Recommendation?

Accept as is

Comments to the Author(s)

The authors considered all the comments of the reviewer, and I suggest the revised manuscript published as is.

Decision letter (RSOS-190179.R1)

10-Jun-2019

Dear Dr Guo,

I am pleased to inform you that your manuscript entitled " $A\beta$ 42 fibril formation from predominantly oligomeric samples suggests a link between oligomer heterogeneity and fibril polymorphism" is now accepted for publication in Royal Society Open Science.

Kind regards,

Alice Power

Editorial Coordinator

Reviewer comments to Author:

Reviewer: 3

Comments to the Author(s)

The authors considered all the comments of the reviewer, and I suggest the revised manuscript published as is.

Reviewer: 1

Comments to the Author(s)

No further comments.

Reviewer: 2

Comments to the Author(s)

I have only one comment, this paper from the references has pages: "New Mechanism of Amyloid Fibril Formation." Galzitskaya O. *Curr Protein Pept Sci.* 2019;20(6):630-640.

Appendix A

Response to reviewer's comments

We thank the reviewers for their review and comments. The new references suggested and some details pointed out by the reviewers help greatly to improve this manuscript. We have revised our manuscript accordingly. Detailed below are our point-by-point responses.

Reviewers' Comments to Author:

Reviewer: 1

Comments to the Author(s)

In this work, the authors studied the oligomer heterogeneity of A β 42 and the kinetics of oligomer and fibril formation. This work is important since A β is a well-known amyloid protein but its oligomerization and fibrillization are still not fully understood. Here, the authors used a combination of practical techniques, including ultrafiltration, TEM, and fluorescence methods, and observed a wide range of oligomer sizes with distinct fibril formation kinetics. The study was well executed and the data support the conclusions.

Minor comments:

1). Why do the smaller oligomers form fibrils faster than the larger ones? Perhaps provide some rationale or speculation in the discussion.

Response: We have added more discussion on why the smaller oligomers form fibrils faster than the larger ones. The relevant text is copied below for reviewer's convenience

On page 6, paragraph 2:

“Two main factors contribute to the apparent fibril growth rate: the rate of fibril elongation and the concentration of fibril nuclei. Because the same A β 42 protein was used, the rate of fibril elongation is likely similar for different filtrates. Therefore, we speculate that the high fibril growth rate for the 100 kD sample is mainly due to higher concentrations of fibril nuclei than the 1000 kD and 0.2 μ m filtrates.”

On page 7, paragraph 3:

“Therefore, our model suggests that A β 42 samples of smaller oligomers would form a higher concentration of fibril nuclei, consistent with the higher apparent fibril growth rate (Figures 3 and 4). The shorter lag time for the fibrillization of smaller oligomers is likely due to a faster conversion rate from oligomers to fibril nuclei.”

2). Page 7, line 4, could the authors provide a reference for “a separate study of the Guo laboratory”?

Response: The mentioned study is still ongoing and the data are not yet published. In the revised text, we added “unpublished data” to clarify this.

3). *Figure 1, would the size distribution of the oligomers be different with longer incubation times than 10-20 minutes after the buffer exchange?*

Response: We did not investigate the potential changes in oligomer size distribution upon further incubation at 4°C. The main objective of this work is to capture the size distribution at the first moment of A β aggregation. Therefore, every effort is given to minimize the delay from solubilization of A β in an aqueous buffer to subsequent experiments, including ultrafiltration, concentration determination, and aggregation. The aggregation experiments suggest that the oligomers formed at the beginning of the aggregation do not re-equilibrate after filtration. Otherwise, we would not have observed the different fibrillization kinetics from different filtrates. The lag time of fibrillization at 37°C is on the order of hours, suggesting that the oligomers are stable for at least hours, if not longer. We believe that this is an important point that may deserve to be further investigated in the future, but it does not have any impact on the conclusions of this study.

4). *Figure 2, what were the concentrations of the original unfiltered samples?*

Response: The concentrations of A β in CG buffer were all at 50 μ M. After buffer exchange, all samples were used for filtration experiments, and the concentration of the unfiltered sample was not measured. The point of Figure 2 is to show that the concentrations of the larger and smaller oligomers vary greatly from sample to sample, and from batch to batch. And this conclusion is not affected by the information on the concentration of the unfiltered sample. We agree that this might be a useful data point, if available, to see if the size distribution of oligomers is related to the concentration of the starting unfiltered sample and will be considered in future studies. And we thank the reviewer for pointing this out.

Reviewer: 2

Comments to the Author(s)

The relationship between oligomers and amyloid fibrils in the aggregation process has been clarified in these papers (PMID: 27500912; 27226198; 28671122; 29338255; 30480261). The authors should add this information in the introduction.

Moreover, using electron microscopy and others methods it has been shown for Abeta peptides and their fragments and also for insulin and the peptide fragments from the BGL2P protein that amyloid fibrils consist of oligomers (PMID: 27500912; 27226198; 28671122; 29338255; 30480261; 30686252).

The peak of sizes on Fig. 1b (8 nm) corresponds to the size of oligomers that contribute to the formation of the amyloid fibrils for Abeta40 and Abeta42 (see these papers PMID: 28671122; 27016282). According to the results of this manuscript the amyloid fibrils also are constructed from the oligomers.

It should be underline that folding nucleus for globular proteins and for formation of fibrils are different (PMID: 24404849), because as has been demonstrated the primary folding nucleus for Abeta40 consists of 2 monomers and for Abeta42 - 3 monomers (see the papers PMID: 27016282; 28671122), but folding nucleus for Abeta peptide is different (for comparison please see this paper PMID: 15632297).

The authors should accent some points in the paper taking into account this new information.

Response: We thank the reviewer for bringing this new information to our attention. We have revised our manuscript taking into account these references suggested by the reviewer.

In the first paragraph of Introduction (page 3), we added the references to ring-like oligomers as building blocks of amyloid fibrils (refs 11-15).

“Some studies suggest that oligomers eventually are converted to amyloid fibrils [9,10] or form the building blocks of amyloid fibrils [11–15], and other studies show that the amyloid fibrils can also promote the formation of oligomers [16].”

In the second paragraph of Introduction (page 3), we added references 32 and 33 and a new sentence to refer readers to these studies.

“There are also studies suggesting that A β forms ring-like oligomers, which form the building blocks of amyloid fibrils [15,32,33].”

In Results and Discussion, we added new references for the oligomer formation of A β 40 and A β 42 (page 4, paragraph 2). References 13 and 34 are added to this part of discussion.

“Previously, fast oligomerization has also been observed for both A β 40 and A β 42 [9,13,34,35,38,39], suggesting that oligomerization may be the first step of A β aggregation.”

We removed a sentence that directly compares protein folding with fibril nucleation to avoid confusion. The removed sentence is “*We envision that the initial oligomerization process is similar to the formation of transition state in the protein folding*” on page 8, first paragraph. As a result, the related discussion on comparison of protein folding nucleus and fibril nucleus was also removed.

Reviewer: 3

Comments to the Author(s)

Aggregation of Abeta peptide has been considered to be closely related with the pathology of Alzheimer’s disease, however the stochastic characteristic of oligomer and fibril formation of Abeta makes it particularly challenging for reliable investigation of the fundamental mechanistic roles of the peptide in AD pathology. Xue et al. in this manuscript reported an elegant study of the potential correlation of oligomer heterogeneity and fibril polymorphism of Abeta. They performed ultrafiltration with variant cutoff size to separate Abeta oligomers with different size. The aggregation kinetics of these oligomers were monitored and compared, showing that smaller oligomers surprisingly forms fibrils with a faster rate. The difference of EGCG solubilization of fibrils from oligomers with distinct size further indicate the structural variances of the fibrils.

The experiments were carried out thoroughly to confirm the aggregation difference observed, and the data was carefully analyzed and well discussed. The results provide new mechanistic insight into the polymorphism nature of the amyloid structures. The manuscript is suitable for publication, after considering some minor points addressed below:

1. *“ADDL” was mentioned in the second paragraph of page 4. The full meaning of this abbreviation need to be provided.*

Response: ADDL is the abbreviation of A β -derived diffusible ligand, and we have now added this to both the main text and the abbreviations list.

2. *The authors reported a surprising fast oligomer formation rate of Abeta42 sample. Indeed, the fast formation of oligomers of Abeta42/42 has been realized in literatures (e.g., fast oligomer formation within hours for Abet40, a moderate aggregation prone one compared to Abeta42, Elbassal et al., JPCC, 2017, 20007). The authors may include more relevant literature reports in the discussion.*

Response: We thank the reviewer for highlighting this point. We have added more discussion on this and have added the references of Elbassal et al. 2017 (ref. 38) and others on page 4, paragraph 2. This part of the text is copied below for reviewer’s convenience:

“Previously, fast oligomerization has also been observed for both A β 40 and A β 42 [9,13,34,35,38,39], suggesting that oligomerization may be the first step of A β aggregation.”